# An Information-Theoretic Criterion for Efficient Data Synthesis

Hanyu Li [1]   Zhengqi Sun [2]   Xiaotie Deng [1]

## Abstract

Synthetic data becomes crucial for large language model training, but its effectiveness is highly inconsistent. We provide an information-theoretic account of this inconsistency: synthetic data improves a model only when the generation-training loop is information-open, i.e., shaped by external signals (verifiers, environments, or rubrics) that inject task-relevant information beyond the model's current distribution. When the loop is information-closed (relying on the model's own outputs without such signals), the data processing inequality ensures that task-relevant information can only decrease, making collapse a predicted outcome. Among information-open pipelines, both efficiency and generalization hinge on the meta-level of supervision: a coarser signal such as binary correctness treats all acceptable outputs as equivalent, so the behavior it teaches is not tied to any particular domain or surface form and generalizes naturally across tasks and domains. These observations lead to a guiding thesis: learning preferentially converges to the most information-efficient signal component available, which accelerates learning when that component is the intended one, but causes reward hacking when a spurious pattern happens to be simpler.

## 1. Introduction

Scaling laws have driven steady progress in large language models (LLMs), with performance improving predictably as model size, compute, and training data grow (Kaplan et al., 2020). This paradigm, however, rests on a premise that is becoming increasingly fragile: the continued availability of abundant, high-quality real-world text. Recent analyses suggest that such data may be approaching practical lim-

its (Villalobos et al., 2024). At the same time, as LLMs are pushed toward long-horizon reasoning, multi-step planning, and agentic interaction, the most effective forms of supervision increasingly rely on explicit constraints, feedback, and verification—going well beyond what generic text corpora can provide. These two trends—data scarcity and the demand for richer supervision—have made synthetic data a central ingredient in modern training pipelines.

Yet synthetic data remains a double-edged sword. On one hand, generated supervision has produced substantial gains: verifier-guided synthesis reaches olympiad-level geometry (Trinh et al., 2024), and reinforcement learning from verifiable rewards has driven major reasoning breakthroughs (Guo et al., 2025; Zheng et al., 2024). On the other hand, iterative self-training—where models are trained on their own outputs—often leads to distribution collapse and progressive capability degradation (Shumailov et al., 2024; Alemohammad et al., 2024; Gerstgrasser et al., 2024). Despite numerous empirical mitigation techniques, a principled account of when and why synthetic data helps or hurts is still lacking. This motivates a concrete question: *under what conditions does synthetic data yield sustained improvements, and when does it induce degradation?*

We approach this question through an information-theoretic lens, anchored in the data processing inequality (DPI). The DPI states a simple but powerful fact: no computation on observed data can create new information about the underlying source. Applied to self-training, the implication is immediate. When a model trains on samples drawn from its own distribution and no additional task-relevant signal enters the loop, the pipeline is *information-closed*: each iteration can only preserve or lose information about the true task. In practice, finite sampling and optimization errors make the loss strict, so the model's knowledge of the task degrades over iterations. Model collapse is thus not an anomaly—it is the predicted outcome of a closed information loop.

This DPI argument, however, is too pessimistic on its own: it cannot explain why synthetic data so often works spectacularly well. The resolution is that successful pipelines are not actually closed systems. They incorporate *external signals*—verifiers, rubrics, environments, or fixed teacher models—that carry task-relevant information independent of the current model state. Once we account for these sig-

[1]CFCS, School of Computer Science, Peking University, Beijing, China [2]Department of Information Management, Peking University, Beijing, China. Correspondence to: Hanyu Li <lhydave@pku.edu.cn>.

*Proceedings of the 43rd International Conference on Machine Learning*, Seoul, South Korea. PMLR 306, 2026. Copyright 2026 by the author(s).

nals, the DPI argument sharpens rather than breaks: the pessimistic monotonicity holds only when such signals are absent. When an external signal is present, the relevant information bound includes not just the training data but also what the signal contributes beyond it. The question then shifts from "can synthetic data add information?" to "how much task-relevant information does the external signal carry beyond what the model already knows?" Sustained improvement is possible precisely when this quantity remains positive—that is, when the loop stays *information-open*.

Given that a pipeline is information-open, a natural follow-up is: *how efficiently* does it inject information? We argue that a key determinant is what we call the *meta-level* of the supervision signal—the granularity at which it distinguishes outputs. A low meta-level signal specifies a particular target (e.g., a single reference answer); a high meta-level signal specifies only a coarse property (e.g., "is the answer correct?") and treats all outputs satisfying that property as interchangeable. Because a high meta-level signal collapses many surface-level differences into a single judgment, each unit of supervision constrains a much larger region of the output space, making learning markedly more sample-efficient.

A natural implication, and a guiding thesis of this paper, is that *learning preferentially converges to the most information-efficient signal component available* in the training data. This thesis cuts both ways: when the intended signal is the most efficient one, it explains why high meta-level supervision produces rapid learning and strong cross-domain generalization; when a spurious signal happens to operate at a coarser granularity than the intended one, the same dynamic produces reward hacking.

**Conflict of Interest Disclosure.** The authors declare no financial conflicts of interest related to the work presented in this paper.

## 2. Case Studies on Synthetic Data

We begin by defining precisely what we mean by synthetic data and identifying three recurring empirical patterns that any theoretical account must explain.

We use *synthetic* in a mechanism-based sense: supervision is synthetic if it is produced by an automated pipeline (often an LLM generator plus tools) and used for training. Under this definition, "synthetic" does *not* mean "fake": it only indicates that the supervision originates from an automated procedure rather than direct human annotation.

In much of the literature, "synthetic data" is used roughly synonymously with *model-generated text*. Our definition is broader in *what* counts as supervision (not only instruction–response pairs but also preferences, rewards, verifier feed-

back, and retained trajectories). Concretely, even when prompts (and sometimes final answers) come from real datasets, reinforcement learning with verifiable rewards (RLVR) can still generate intermediate trajectories that are retained only if they pass verifiable checks; those retained trajectories function as *synthetic supervision* in our sense. With this terminology, we summarize three recurring patterns below: closed-loop self-training (failure), hard verifiers (success I), and soft but stable references (success II).

**Failure (closed-loop self-training).** When training primarily recycles the model's own generations *without* a persistent external signal, the loop tends to contract the output distribution: tail patterns are under-sampled and errors become self-reinforcing (Shumailov et al., 2024; Alemohammad et al., 2024). The effect is strongest when synthetic data progressively *replaces* real data, whereas mixing real data each iteration can mitigate collapse (Gerstgrasser et al., 2024). Synthetic-heavy regimes can also distort scaling behavior by weakening long-tail modeling (Dohmatob et al., 2024).

**Success I (hard, verifiable signals).** A high-reliability path is *generate-and-verify*: candidates are filtered, ranked, or rewarded by an external procedure (proof checker, compiler/tests, benchmark harness), via rejection sampling, best-of-$N$, or RLVR (Guo et al., 2025; Duo et al., 2026). The headline outcome is *qualitative performance jumps anchored by checkable signals*: in math, verifier-guided search and distillation can reach medal-level performance (e.g., IMO 2024 silver-medal standard) (Hubert et al., 2025; AlphaProof & AlphaGeometry teams, 2024), and AlphaGeometry-style constrained synthesis reaches gold-medalist-level geometry solving (Trinh et al., 2024; Chervonyi et al., 2025). In formal proving, similar recipes scale to strong benchmark performance under Lean verification (Ren et al., 2025; Chen et al., 2025b;a). In discovery loops, formal/executable targets let systems like Fun-Search, LegoNE, and AlphaEvolve report algorithmic improvements that are *certified* by solvers or proof obligations (including guarantees that surpass prior best known human-designed ones) (Romera-Paredes et al., 2024; Li et al., 2025; Novikov et al., 2025).

In code and systems, execution provides the verifier: AlphaCode and OpenCodeInterpreter rely on tests/runtime feedback (Li et al., 2022; Zheng et al., 2024), and agentic pipelines evaluate artifacts in containerized environments (DeepSeek-AI et al., 2025; Xiaomi LLM-Core Team et al., 2026). The same template supports optimization loops in SAT solving and code evolution (Sun et al., 2024; 2025) and extends to kernel performance engineering with correctness checks plus runtime selection (Chen et al., 2025c;

Ouyang et al., 2025).

**Success II (soft but stable references).** For objectives without crisp correctness predicates, stability comes from *holding the reference fixed*: teacher models, preference/rubric constraints, constitutions, or evaluation harnesses that do not co-move freely with the student. Examples include distillation and preference-based alignment (Hinton et al., 2015; Gudibande et al., 2024; Tunstall et al., 2024), on-policy distillation with teacher-side judging (Xiaomi LLM-Core Team et al., 2026; Agarwal et al., 2024; Lu, 2025), fixed-principle self-improvement (Bai et al., 2022), staged filtering pipelines (Gudibande et al., 2024), and iterative search under a fixed evaluator (Yuan et al., 2024).

# 3. A Theoretical Analysis through Data Processing Inequality

As is discussed in the case studies, some synthetic-data pipelines behave like a closed system that only recycles the model's own distribution, while others remain open because they are persistently shaped by signals not determined by the model itself. This section makes that distinction precise using the data processing inequality (DPI), and then states a practical criterion for when synthetic data can be effective.

Our goal is not to deny DPI, but to clarify the modeling boundary: DPI applies to an *information-closed* Markov abstraction. Synthetic data can be effective only when the actual pipeline is *not* information-closed, i.e., when it contains additional variables (external signals, verifiers, fixed references, environments) that are correlated with the underlying source $X$ and therefore cannot be absorbed into a single fixed channel.

## 3.1. Markov Chain Formalization

Let $X$ denote the underlying source of structure we ultimately care about (e.g., the task distribution that induces correctness), $D$ denote the finite observations used for learning (e.g., a training dataset), and $Z$ denote the learned model state (e.g., parameters).

We model training as a (possibly randomized) procedure that maps $D$ to $Z$. To keep the role of stochasticity explicit, introduce an auxiliary random variable $R$ that aggregates all internal randomness used by the algorithm (initialization, minibatch sampling, optimizer noise, decoding randomness, etc.). The training procedure can then be written as

$$Z = f_{\text{train}}(D, R).$$

The *information-closed* assumption is that the procedure has no additional side information about $X$ beyond what is already in $D$; equivalently,

$$P(Z \mid X, D) = P(Z \mid D),$$

so that $X \to D \to Z$ forms a Markov chain.

Under this assumption, DPI yields

$$I(X; Z) \leq I(X; D),$$

where $I(\cdot; \cdot)$ is Shannon mutual information. A compact proof follows from the chain rule:

$$\begin{aligned} I(X; D, Z) &= I(X; D) + I(X; Z \mid D) \\ &= I(X; Z) + I(X; D \mid Z), \end{aligned}$$

and Markovness implies $I(X; Z \mid D) = 0$; hence $I(X; Z) = I(X; D) - I(X; D \mid Z) \leq I(X; D)$.

Finally, it is useful to separate "randomness" from "information." Internal randomness $R$ can change optimization dynamics and exploration, but it does not by itself increase information about $X$ unless it is coupled to some variable correlated with $X$. Formally, if $R$ carries no information about $X$ beyond $D$, then $I(X; D, R) = I(X; D)$ and DPI applied to $X \to (D, R) \to Z$ still gives the same upper bound.

## 3.2. Failure of Self-Training Loops

Consider a purely self-training loop. At iteration $t$, a model state $Z_t$ generates synthetic supervision $D_t^{\text{syn}}$ via a (possibly stochastic) generation procedure $f_{\text{gen}}$ (the "synthetic data" in the mechanism-based sense of this paper), and the next model state is obtained by training on that synthetic supervision:

$$D_t^{\text{syn}} \sim f_{\text{gen}}(Z_t), \qquad Z_{t+1} = f_{\text{train}}(D_t^{\text{syn}}, R_t).$$

If the pipeline introduces no persistent external signal (no fresh human data, no environment feedback, no fixed verifier/reference that is independent of the current model state), then the iteration is information-closed in the sense that all variables downstream of $Z_t$ are generated from $Z_t$ and internal randomness. In that case we have the Markov chain

$$X \to Z_t \to D_t^{\text{syn}} \to Z_{t+1}.$$

Applying DPI to this chain yields a monotonicity statement:

$$I(X; Z_{t+1}) \leq I(X; Z_t) \qquad \text{for all } t.$$

This captures the core limitation of closed-loop self-training: when synthetic data is sampled from the model's own distribution and then fed back without any independent signal, the loop cannot systematically increase the model's information about $X$. In principle, the best possible outcome is

information preservation; in practice, finite sampling, capacity limits, and approximation errors typically make the inequality strict, so performance degrades over iterations.

As a concrete instance of this phenomenon beyond the well-studied model collapse setting, we observed the following in ongoing work on instruction-following evaluation. A judge model was trained to assess whether LLM responses satisfy user-specified rubrics (e.g., "is the reply concise?", "does the response follow the requested format?"). In one training variant, the judge was asked to *infer* implicit rubrics from the conversation context and then evaluate compliance—without any external ground-truth rubrics to anchor its judgments. When this judge was trained iteratively (using its own previous outputs as training signal for the next iteration, each time in a fresh context), its acceptance rate decreased monotonically across iterations: it became progressively more critical of outputs, eventually rejecting nearly everything. The degradation was not because the evaluated outputs worsened—they were held constant—but because the judge's internal criteria drifted without external correction. Each iteration appeared locally reasonable (the judge produced plausible critiques), yet the cumulative effect was systematic quality collapse invisible from within the loop. This is precisely the closed-loop monotonicity predicted above: without a stable external anchor, the "signal" becomes a function of the model's own evolving state.

### 3.3. Information-Openness and Effective Synthetic Data

The preceding limitation is not a special property of language models; it is a direct consequence of modeling the loop as information-closed. The key question is therefore not whether data is "synthetic," but whether the pipeline is *information-open* with respect to $X$.

Modern successful pipelines typically introduce *external signals* that shape either the generation stage or the training stage. Here an "external signal" means any random variable $S$ that (i) influences which synthetic outputs are retained or amplified, and (ii) is not itself determined solely by the model's current output distribution. Examples include executable verifiers, environments, frozen judges or rubrics, and fixed references that do not freely co-move with the current model. Formally, we call a pipeline *information-open* if $I(X; S \mid D) > 0$, i.e., the external signal carries information about $X$ beyond what is already contained in $D$.

Once such an $S$ exists, treating the whole pipeline as a single fixed channel $P(Z \mid D)$ becomes misleading: the learned model is no longer a function of $D$ alone. The correct information-theoretic object is the augmented observation $(D, S)$. In this view, synthetic data can be effective only insofar as $S$ provides task-relevant constraints or feedback that are correlated with $X$ and remain stable enough to act as a signal across iterations.

### 3.4. Refining DPI via External Signals

We now refine the DPI argument by accounting for external signals: we do not violate DPI; rather, the relevant observation is the augmented pair $(D, S)$ rather than $D$ alone, so the pessimistic closed-loop DPI monotonicity need not apply.

Let $S$ denote an external signal used by the pipeline, which is a random variable that may depend on $X$ and $D$. We model training as

$$Z = f_{\text{train}}(D, S, R).$$

Then $X \to (D, S) \to Z$ is a Markov chain, and DPI gives

$$I(X; Z) \le I(X; D, S).$$

By the chain rule,

$$I(X; D, S) = I(X; D) + I(X; S \mid D).$$

This yields a simple criterion for effectiveness: a synthetic-data pipeline can improve the model's information about $X$ beyond what is available in $D$ only through the additional conditional mutual information term $I(X; S \mid D)$. Equivalently, synthetic data is effective when the external signal contributes nontrivial information about $X$ that is not already implied by $D$.

For iterative synthetic training, the same idea produces a per-iteration bound. If $Z_{t+1}$ is produced from $(Z_t, S)$ (possibly via intermediate synthetic data), then $X \to (Z_t, S) \to Z_{t+1}$ and

$$I(X; Z_{t+1}) \le I(X; Z_t, S) = I(X; Z_t) + I(X; S \mid Z_t).$$

This clarifies when an iteration can avoid closed-loop monotonicity: without $S$, the additional capacity term $I(X; S \mid Z_t)$ vanishes and the bound reduces to the closed-loop monotonicity $I(X; Z_{t+1}) \le I(X; Z_t)$; with an $S$ that remains meaningfully correlated with $X$ conditioned on $Z_t$, the loop can, in principle, introduce new task-relevant constraints and sustain improvement.

This also clarifies the role of stochasticity. Random sampling does not by itself inject information about $X$—it merely generates a diverse pool of candidate outputs. The external signal then acts as a filter on this pool, retaining or reinforcing candidates that happen to align with the task. Neither component alone suffices: without randomness, the model has no diversity to select from; without the signal, diverse candidates cannot be distinguished. Effective synthetic pipelines thus operate as an explore-then-select loop, where stochasticity provides exploration and $S$ provides selection pressure.

The per-iteration bound is also consistent with the observation that the *structure* of noise matters far more than its magnitude. Unbiased noise (random disagreements that are independent across samples) cancels in expectation across gradient updates and does not systematically corrupt learning. By contrast, systematic bias accumulates coherently; and if the signal co-evolves with the student model (e.g., a judge that drifts without calibration, as in Section 3.2), it ceases to be genuinely external and the loop effectively recloses. We return to this asymmetry with detailed empirical evidence in Section 4.

A striking engineering validation of this stability requirement comes from Anthropic's generator-evaluator harness for long-running agentic applications (Rajasekaran, 2026), which architecturally separates a generator agent (that proposes solutions) from a fixed evaluator agent (that judges quality). The evaluator is deliberately held constant and does not adapt to the generator's outputs; it functions as a stable external signal in precisely the sense of Section 3.3. This design yielded large gains over single-agent approaches where the same model both generates and evaluates—exactly as the framework predicts: a co-moving evaluator closes the loop and triggers degradation, while a fixed evaluator keeps the loop information-open.

### 3.5. How External Signals Enter Training

To make the external-signal framework concrete, we compare three representative methods that form a natural progression in how much they rely on an external signal: supervised fine-tuning (SFT), rejection-sampling fine-tuning (RFT), and reinforcement learning with verifiable rewards (RLVR, often implemented with GRPO-style updates (Shao et al., 2024)).

In standard SFT, we optimize log-likelihood on a fixed dataset:

$$\max_\theta \ \mathbb{E}_{(x,y)\sim q_{\text{data}}}\big[\log \pi_\theta(y \mid x)\big].$$

Each training pair $(x, y^\star)$ contributes a gradient $g_{\text{SFT}} = \nabla_\theta \log \pi_\theta(y^\star \mid x)$ that simply increases the probability of a provided target. There is no external signal beyond the dataset itself.

RFT introduces an external signal through *filtering*. The current model proposes candidates $y \sim \pi_{\theta_t}(\cdot \mid x)$, and an acceptance test $a(x, y; S) \in \{0, 1\}$ (e.g., a correctness check) retains only those that pass. SFT is then run on the retained samples, whose distribution $\pi_{\theta_t}^{\text{acc}}(y \mid x, S) \propto \pi_{\theta_t}(y \mid x) \, a(x, y; S)$ is jointly shaped by the model proposal and the signal. The signal reshapes *which samples become training data*.

RLVR introduces the external signal directly into the *gradient*. A verifier returns a reward $r(x, y; S)$ for model-generated outputs, and the policy gradient

$$g_{\text{RLVR}}(\theta; x) \propto \mathbb{E}_{y\sim\pi_\theta(\cdot|x)}[A(x, y; S) \, \nabla_\theta \log \pi_\theta(y \mid x)],$$

where $A(x, y; S) \coloneqq r(x, y; S) - b(x)$ is an advantage relative to a baseline, directly weights each sample's contribution by the signal. Outputs with above-baseline reward get positive weight; below-baseline outputs get negative weight. Unlike SFT, RLVR does not specify any particular target output—it selectively reinforces or suppresses model-generated behaviors based on the external signal alone.

The SFT–RFT–RLVR progression above uses hard verifiable signals, but the same $(D, S)$ formalism applies when $S$ is a soft reference—such as a fixed rubric evaluated by an LLM judge (Section 4) or a held-out evaluator that does not co-adapt with the student. The critical requirement is not hardness of verification but *stability*: $S$ must not freely co-move with $Z_t$, so that $I(X; S \mid Z_t)$ remains positive across iterations.

## 4. Information Injection Efficiency

The previous section establishes an information-openness criterion: synthetic data can improve a model only when an external signal injects task-relevant information that the model cannot generate on its own. This criterion, while precise, is largely a formalization of an intuitive observation—closed loops degrade, open loops can improve. The more substantive question is what happens *among* information-open pipelines: given that some external signal exists, why do different methods exhibit orders-of-magnitude differences in sample efficiency and generalization? Some pipelines require massive volumes of synthetic supervision to yield marginal gains, while others trigger broad, cross-task behavioral changes with comparatively little data. This section develops a quantitative framework that explains this variance and identifies the structural property of the supervision signal that determines efficiency.

### 4.1. Meta-Level of Information Injection

We define *information injection efficiency* as the fraction of supervision information that is directed at task-relevant distinctions, rather than at irrelevant within-class details. Intuitively, a synthetic-data step is efficient if it eliminates much of the model's uncertainty about what it should output, rather than merely adding more surface-form examples. To connect this intuition to information theory with minimal notation, we fix a prompt random variable $Q$ and an output random variable $Y$ taking values in an output space $\mathcal{Y}$ (an entire model response). A synthetic-data pipeline produces a supervision signal $S$ about the pair $(Q, Y)$ (e.g., accept/reject, a score, a preference). The uncertainty reduction caused by one such supervised interaction is $I(Y; S \mid Q)$—the number of bits by which $S$ reduces the model's uncertainty over its

outputs.

The remaining question is what makes this quantity large or small. The key idea of *meta-level* can be stated informally: different supervision signals care about different aspects of an output. Some signals care about a very specific surface form (a unique target string), while others only care about a coarse property (e.g., "is it acceptable?", "does it compile?"). A higher meta-level signal *ignores* many superficial differences between outputs and only distinguishes a small number of behaviorally meaningful categories. When supervision is higher meta-level, each labeled example can constrain a larger portion of the output space because it rules out whole categories at once, instead of identifying a single preferred surface form.

A simple quantitative example makes this concrete. Fix a prompt $q$ and suppose there are $M$ different answers that are all equally acceptable under an external criterion (e.g., multiple correct proofs, multiple valid patches). Denote this acceptable set by $\mathcal{A} \subseteq \mathcal{Y}$ with $|\mathcal{A}| = M$. Assume for clarity that the model's uncertainty within $\mathcal{A}$ is roughly uniform.[1]

**High meta-level supervision (class-level signal).** A high meta-level pipeline provides a coarse signal that only cares about whether the output falls inside the acceptable set:

$$S_{\text{high}}(q, y) = \mathbf{1}\{y \in \mathcal{A}\}.$$

This signal deliberately treats all $M$ acceptable answers as equivalent: it does not distinguish between different members of $\mathcal{A}$. The uncertainty reduction contributed by observing this signal is

$$I(Y; S_{\text{high}} \mid Q=q) = H(S_{\text{high}} \mid Q=q),$$

because $S_{\text{high}}$ is a deterministic function of $Y$ given $q$. If the model is unsure whether it will land inside $\mathcal{A}$ (accept) or outside (reject), then $H(S_{\text{high}} \mid Q=q)$ can be close to one bit, meaning each feedback instance can remove close to one bit of uncertainty by collapsing the space into just two categories: acceptable vs. unacceptable. Crucially, this signal never spends any information budget telling the model *which* acceptable answer to prefer; it only tells the model to move probability mass into the acceptable region.

**Low meta-level supervision (instance-level identification).** A low meta-level pipeline instead provides a *specific target* $y^\star \in \mathcal{A}$ and trains the model to reproduce that particular surface form (e.g., a single canonical reference answer):

$$S_{\text{low}}(q, y) = \mathbf{1}\{y = y^\star\}.$$

This signal is much finer: it distinguishes one specific acceptable output from the other $M - 1$ acceptable outputs that are, from the external criterion's perspective, equally good. Under the uniform-within-$\mathcal{A}$ assumption, the probability that the model outputs exactly $y^\star$ is about $1/M$, hence the information gain of this indicator is approximately

$$H(S_{\text{low}} \mid Q=q) \approx h_2(1/M),$$

where $h_2(x) = -x \log x - (1-x) \log(1-x)$ is binary entropy. For large $M$ (which is typical in practice), $h_2(1/M)$ is very close to zero, so each observation of $S_{\text{low}}$ provides a very weak information signal about the model's output space: almost always $S_{\text{low}} = 0$, which tells the learner little about what to do next.

More importantly, reproducing $y^\star$ forces the learner to resolve an additional identification burden inside the acceptable set. If the acceptable answers are all treated as different targets, then "being correct" is not enough; the learner must also identify *which* member of the acceptable set is desired. The maximum number of bits needed to identify a particular member among $M$ acceptable ones is $\log M$. This can be seen directly from the within-class entropy: under uniformity,

$$H(Y \mid Q=q, Y \in \mathcal{A}) = \log M.$$

Instance-level supervision implicitly attempts to drive this within-class entropy toward zero by selecting a single representative, while high meta-level supervision does not. Therefore, when $M$ is large, a low meta-level objective can waste a substantial portion of supervision capacity on distinctions that are irrelevant to acceptability, while the high meta-level objective concentrates only on the coarse distinction that matters.

In verifiable domains such as mathematics, this is the key difference between SFT and RLVR. SFT imitates a specific reference $y^\star$, spending capacity on identifying which particular acceptable answer to produce—a distinction irrelevant to the task criterion. RLVR uses verifiable correctness (membership in $\mathcal{A}$), treating all correct solutions as equivalent and training the model to reach *any* correct answer. This ties learning to an invariant criterion, letting the model discover and reuse any verifier-satisfying strategy, which typically yields stronger out-of-distribution robustness.

A direct validation of this prediction is JudgeRLVR (Duo et al., 2026), which trains a judge model on mathematics using only binary correctness as supervision—the signal is purely whether the response answers the query correctly, without any domain-specific rubric or reference solution. Because this signal treats all correct solutions as equivalent and does not depend on the domain being mathematics, the trained judge transfers to all other domains (code, logic, general reasoning) without further supervision, outperforming

---

[1]Uniformity is not necessary for the argument; it only makes the bound explicit. In general the within-class entropy is at most $\log M$.

judges trained with domain-specific rewards. The mechanism is exactly what the meta-level analysis predicts: binary correctness is the coarsest task-relevant signal, so every bit of learned judgment generalizes across domains rather than being spent on domain-specific distinctions.

*In this sense, generalization is precisely the ability to correctly "forget" irrelevant differences—to learn at a higher meta-level.*

## 4.2. Formalizing the Efficiency Gap

The intuitive comparison above can be generalized into a decomposition theorem that applies to any supervision signal and any task-relevant distinction.

**Definition 4.1** (Task-Relevant Partition). A **task-relevant partition** $\pi = \{C_1, \ldots, C_K\}$ of $\mathcal{Y}$ is a partition such that the external evaluation criterion assigns the same judgment to all elements within the same block. The induced **quotient variable** $[Y]_\pi \in \{1, \ldots, K\}$ is defined by $[Y]_\pi = k$ whenever $Y \in C_k$. A supervision signal $S = \sigma(Q, Y)$ is $\pi$-**measurable** if $S$ depends on $Y$ only through $[Y]_\pi$.

The accept/reject partition of the preceding example ($K = 2$, $C_1 = \mathcal{A}$, $C_2 = \mathcal{Y} \setminus \mathcal{A}$) is a special case. In general, $\pi$ can encode any evaluation structure: multi-level rubric scores, partial-credit grading, or tiered quality judgments.

**Theorem 4.2** (Decomposition of Supervision Information). *For any supervision signal $S$ and task-relevant partition $\pi$,*

$$I(Y; S \mid Q) = \underbrace{I\big([Y]_\pi; S \mid Q\big)}_{\text{task-relevant gain}} + \underbrace{I\big(Y; S \mid [Y]_\pi, Q\big)}_{\text{within-class gain}}.$$
(1)

*Assuming $I(Y; S \mid Q) > 0$, we define the **task-relevant efficiency** $\eta_\pi(S) := I([Y]_\pi; S \mid Q) / I(Y; S \mid Q)$. Then: (i) $\eta_\pi(S) \in [0, 1]$; (ii) $\eta_\pi(S) = 1$ iff $Y \perp\!\!\!\perp S \mid [Y]_\pi, Q$; (iii) every $\pi$-measurable signal satisfies $\eta_\pi(S) = 1$.*

*Proof.* Since $[Y]_\pi$ is a deterministic function of $Y$, the chain rule gives $I(Y; S \mid Q) = I([Y]_\pi; S \mid Q) + I(Y; S \mid [Y]_\pi, Q)$. Both terms are nonnegative. Part (iii): if $S = g(Q, [Y]_\pi)$, then $I(Y; S \mid [Y]_\pi, Q) = 0$, giving $\eta_\pi = 1$. $\square$

The decomposition formalizes the intuition from the preceding example: the total information injected by any signal splits cleanly into a task-relevant component (resolving which equivalence class the output belongs to) and a within-class component (resolving which specific element within that class). The SFT-vs-RLVR comparison of Section 4.1 is a direct instance: RLVR's binary verifier is $\pi$-measurable ($\eta_\pi = 1$), while SFT's instance-level target spends $\log M$ additional bits on within-class identification—capacity that is irrelevant to the task criterion and grows without bound as the number of acceptable solutions $M$ increases.

A natural implication, consistent with known simplicity biases in gradient-based learning, is that learning will preferentially converge to whichever signal component is most information-efficient to exploit. When the training signal contains multiple learnable patterns at different granularities, the model does not choose among them based on task relevance—it converges to the pattern that provides the largest per-sample gradient signal, which is the one operating at the coarsest partition (highest $\eta$). When the intended task signal happens to be the most efficient one, this is desirable: learning is fast, robust, and generalizes broadly. But when a spurious signal operates at a coarser granularity than the intended one, the model will converge to the spurious pattern first—this is the information-theoretic mechanism underlying reward hacking (see Section 4.4 for a detailed example). This convergence claim is not a formal consequence of the decomposition alone; it additionally relies on the empirical observation that gradient-based learners preferentially exploit patterns with larger per-sample signal, which coarser partitions provide.

The decomposition also reveals why $\pi$-measurable signals are inherently *robust to noise*. Consider rubric-based RL for instruction following, where an LLM judge evaluates whether responses satisfy a set of rubrics (e.g., "Does the response use bullet-point format?", "Is the reply under 200 words?") and a sample receives reward 1 only if all rubrics are judged as satisfied. To measure the judge's intrinsic noise, we scored 1,000 (prompt, response) pairs 16 independent times each. Sample-level consistency—all 16 scorings producing identical verdicts on every rubric—was only 24%. Yet RL training using this judge converged to strong out-of-distribution generalization. The reason connects directly to the theorem: because the reward is $\pi$-measurable (it only distinguishes "all rubrics satisfied" from "not all satisfied"), the noise exists entirely *within* each partition block—different rubric items flip in different directions on different runs, but the between-class signal (satisfies-all vs. not) remains directionally consistent across samples. Random within-class noise cancels in the gradient; the task-relevant signal accumulates. This is a second advantage of high meta-level signals beyond efficiency: they are structurally immune to noise that lacks systematic directionality at the class level.

## 4.3. Application: Why Diversity Beats Volume

The partition framework also explains a widely observed empirical pattern: data *diversity* often matters more than data *volume* for generalization. A small but diverse dataset can outperform a much larger but narrow one. The mechanism is straightforward once we think in terms of partition coverage.

Given a task-relevant partition $\pi = \{C_1, \ldots, C_K\}$ and a $\pi$-measurable signal $S$, consider what happens when the

same (prompt, output) pair is observed repeatedly. Since $S$ is a deterministic function of $(Q, [Y]_\pi)$, repeated observations of the same pair contribute **exactly zero** additional information: letting $S^{(1)}, \ldots, S^{(\ell)}$ denote signals from successive observations of the same pair, $I([Y]_\pi; S^{(\ell+1)} \mid S^{(1)}, \ldots, S^{(\ell)}, Q=q) = 0$ whenever the same output is observed. In contrast, a *new* prompt that exercises an uncovered $\pi$-block contributes up to $H([Y]_\pi \mid Q)$ fresh bits of task-relevant information. The marginal value of a new prompt is therefore bounded below by the gap between the covered and uncovered partition blocks, while the marginal value of a repeated prompt is exactly zero.

This explains why strong alignment has been achieved with only 1,000 diverse examples in prior work: the task-relevant partition for alignment (helpful vs. harmful vs. refusal) has relatively few blocks, so full coverage requires diversity, not volume. Similarly, scaling data-constrained models encounters diminishing returns from repeated data: after the partition is well-covered, additional samples contribute mostly within-class information (if the signal is fine-grained) or zero information (if the same observation is repeated). The practical implication is clear: when designing synthetic-data pipelines, prompt diversity—covering as many distinct $\pi$-blocks as possible—should be prioritized over data volume on already-covered regions.

### 4.4. High-Efficiency Information Injection in Practice

The meta-level view suggests a concrete signature of high-efficiency injection: the supervision signal should induce a constraint that transfers across prompts, domains, and even interaction formats. The following examples illustrate how such transfer emerges from meta-level signals.

A first example is to inject a format that is essentially task-agnostic. DeepSeek-R1-style post-training makes this explicit by rewarding *format compliance* in addition to correctness, e.g., requiring the model to wrap intermediate reasoning and final answers inside tags such as `<think>...</think>` (Guo et al., 2025). Format compliance defines a global constraint on the output space that does not depend on any particular task distribution. Once learned, it manifests as cross-task transfer: the same "output interface" applies to math, coding, and natural-language tasks alike.

The second example is chain-of-thought (CoT) compression, where the injected signal is explicitly meta-level. In mastery-gated CoT compression with sample-level soft penalties, the feedback depends only on the length of current rollouts and is applied only after the model already reaches stable correctness (Li et al., 2026). This signal is agnostic to what the problem is: it does not rely on task identity, domain-specific structure, or even what the model is "doing," as long as the answer remains correct. The result is one-to-

all-domain generalization: training on math alone makes reasoning shorter in code, instruction following, and general QA, and can even reduce the number of turns in agentic settings, while preserving accuracy.

The meta-level framework also explains why certain training failures occur. In ongoing work on training a judge model for math, code, and logic evaluation, we constructed training data by pairing positive examples (correct solutions) with negative examples (incorrect solutions). An inadvertent data-construction choice assigned all positives from one model family (Gemini, which tends to produce longer, more detailed outputs with a distinctive prose style) and all negatives from another (MiMo, which produces shorter, more terse outputs). The intended signal was *correctness*; the spurious signal was *length and style*.

The judge quickly learned the spurious shortcut: rather than evaluating mathematical correctness, it learned to predict length and style. The telltale sign was a divergence between two metrics—the judge's training reward continued to rise (it was getting better at distinguishing long from short outputs), while its actual correctness accuracy on a balanced held-out set plateaued (it was not getting better at judging math). Critically, the reward hacking was invisible from inspecting the model's chain-of-thought—it still appeared to reason carefully about mathematical correctness, showing no obvious change in its reasoning patterns. The failure was only detectable through indirect metrics (validation-set output length stopped increasing, indicating the model was no longer exploring).

Re-balancing the data sources so that length and style were decorrelated from correctness immediately eliminated the hacking and restored accuracy growth. In terms of Theorem 4.2, length and style define a simpler, more easily exploitable pattern than mathematical correctness—the model can detect them from surface statistics without evaluating reasoning—so the spurious signal achieves higher $\eta$ for the *wrong* partition. The model converges to it preferentially because it is more information-efficient to learn.

Finally, meta-level information injection can happen *without* updating parameters, purely by placing the model in a fixed propose–evaluate–select loop. SATLUTION (Sun et al., 2025) targets efficient SAT solvers and LegoNE (Li et al., 2025) aims to discover general algorithms for approximate Nash equilibria with better provable worst-case guarantees. Both systems apply LLMs as explorers and use external verifiers (benchmarks, formal analyzers) to filter and retain only high-quality candidates. The meta-level signal comes from how the verifier *collapses* the search space: many very different proposals are treated as interchangeable once they satisfy the same executable or formal checks. Empirically, SATLUTION evolves solver variants that outperform the human-designed winners of SAT Competition

evaluations, and LegoNE discovers algorithms whose certified worst-case guarantees surpass all previously known human-designed ones.

Taken together, these examples highlight a consistent pattern: strong generalization appears when supervision targets meta-level signals—format compliance, verifier-defined validity, stable process structure, or formally checkable subspaces—so that learning can "forget" irrelevant differences and operate on a simpler quotient space of behaviors.

## 5. Discussion

The guiding thesis of this paper—that learning preferentially converges to the most information-efficient signal component available—reframes a phenomenon often treated as a training failure. Reward hacking is conventionally understood as the model "exploiting" the reward instead of learning the intended behavior. Under our framework, the model is not failing to learn; it is learning exactly what the training signal most efficiently teaches. When a spurious pattern (such as output length or stylistic cues) happens to operate at a coarser granularity than the intended criterion (such as correctness), gradient-based optimization converges to the spurious pattern first, because it is more information-efficient to exploit. This means reward hacking cannot be eliminated simply by training harder or longer—it requires either ensuring that the intended signal is genuinely the most information-efficient component, or decorrelating spurious coarse-grained patterns from the reward.

The framework also identifies a concrete bottleneck for scaling synthetic data: *verification capacity*. Information-open loops require external signals, and the most efficient such signals are high meta-level verifiers—binary correctness checks, executable tests, formal proof obligations. Accordingly, progress is fastest in domains where such verifiers are readily available: mathematics, code, and formal reasoning. In domains where reliable verification is difficult—open-ended generation, nuanced safety judgments, creative tasks—the loop either remains closed or relies on soft references (LLM judges, human rubrics) that are noisier and may co-drift with the model. The practical implication is that expanding the frontier of what can be reliably verified may matter more for capability gains than scaling data or compute alone.

This perspective echoes Sutton's *Bitter Lesson* (Sutton, 2019): general methods that leverage computation ultimately outperform methods that build in human knowledge. In our framing, instance-level supervision—hard-coding particular solutions, formats, or trajectories—is the synthetic-data analogue of "building knowledge in." Meta-level signals, by contrast, specify only what must hold (correctness, format compliance, constraint satisfaction) and let compu-

tation discover how to achieve it—precisely Sutton's prescription to invest in methods that *find and capture* structure rather than in hand-crafted structure itself (Sutton, 2019).

Our framework is primarily qualitative: it identifies structural conditions for effective synthetic data but does not predict, for instance, the sample complexity of a given pipeline. The convergence thesis is supported by our case studies and by known simplicity biases in gradient-based learning, but it is not a formal consequence of the DPI decomposition alone. Finally, the framework takes the external signal as given; analyzing how verifier errors propagate through iterative training, and how to design verifiers that remain reliable as models improve, are important open directions.

## Acknowledgements

This work is supported by the Natural Science Foundation of China (Grant No. 62572010).

## Impact Statement

This paper presents work whose goal is to advance the field of general artificial intelligence. There are many potential societal consequences of our work, none of which we feel must be specifically highlighted here.

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
