# OpenReview forum: "An Information-Theoretic Criterion for Efficient Data Synthesis"
_ICML.cc/2026/Conference — ICML 2026 regular_

### Official Review · Reviewer_XBh4 · 2026-03-09

**Soundness:** 4
**Presentation:** 3
**Significance:** 3
**Originality:** 3
**Overall Recommendation:** 5
**Confidence:** 4

**Summary:**

The paper proposes a formalization of synthetic data for AI model training. It approaches the problem from information theoretical perspective and distinguishes between information-open and information-closed scenarios. In the former, synthetic data can lead to improved models and better generalization, while in the latter the model either stays the same or degrades. The information-open scenarios are further split into verifiable signals and stable references and they are compared from an information theoretic perspective as well. Further, the paper provides some case studies that demonstrate how certain models fit into their framework.

**Compliance With Llm Reviewing Policy:**

Affirmed.

**Key Questions For Authors:**

The paper would benefit from recommendation for synthetic data generation. What are the implications of this way of looking at synthetic data generation? What is the take home message for a practitioner?

**Limitations:**

yes

**Strengths And Weaknesses:**

Strengths:
- The paper proposes a really interesting (and to my eyes novel) distinction between different types of synthetic data, and of ways to think about injecting information into the system
- The paper is well presented, easy to read and easy to understand
- The mathematics appear to be rigorous and present a formal way to look at different types of synthetic data

Weaknesses:
- The paper presents different ways to think about synthetic data in theory, but limited support of it in practice. Several case studies are presented, but they are fairly limited. Only 4 papers cited in section 4.2 which provides some case studies. The paper would significantly benefit from a bigger set of case studies or even a meta-analysis. Alternatively, practical results could be demonstrated through experiments.
- A lot of synthetic data generation is driven with a human in the loop. This ranges from selection/inspection of samples to keep to design of synthetic data generation pipelines (prompts, sampling strategies, multi-stage approaches). It would be great to have a greater discussion of this aspect and where it fits in the proposed framework.
- Trivia: RLVR used before it is defined

---

> ### Author Rebuttal · Authors · 2026-03-27
>
> We thank the reviewer for the encouraging assessment and the concrete suggestions.
>
> **1. On W1: "Only 4 papers cited in section 4.2... would significantly benefit from a bigger set of case studies."** We note that Section 2 already covers a broader set — AlphaGeometry, FunSearch, AlphaCode, DeepSeek-Prover, Constitutional AI, on-policy distillation, and others — organized by pattern (closed-loop failure / hard verifiers / stable references). Section 4.2 focuses on four systems that most cleanly illustrate the high-efficiency regime; we will extend it in revision.
>
> We also note that several systems published *after* our submission date (January 2026) independently instantiate the same principles, which we find encouraging as post-hoc validation:
>
> *Agent-based (no training):* AVO (arXiv:2603.24517, March 2026) uses autonomous agents as evolutionary operators for GPU kernel optimization, evaluated purely on benchmarks — high meta-level verifier, no training. Anthropic's generator-evaluator harness (anthropic.com/engineering/harness-design-long-running-apps, 2026) separates a generator agent from a fixed evaluator, yielding large gains over single-agent approaches — the evaluator does not co-move with the generator, exactly the condition of Section 3.3.
>
> *Training-based:* JudgeRLVR (arXiv:2601.08468) trains a judge on math with verifiable ground truth; the signal is purely whether the response correctly answers the query — task-agnostic and high meta-level (Section 4.1). The trained judge transfers to all other domains without further supervision, outperforming standard RLVR on non-math tasks. From our ongoing work: (i) gpt-oss-120b as reward model in rubric-based RL for instruction following showed only 24% sample-level consistency across 16 scorings of 1,000 (prompt, response) pairs, yet RL converged to strong out-of-distribution (OOD) generalization — random noise cancels (Section 3.4); (ii) a judge trained with all positives from Gemini 3 Pro and negatives from MiMo-V2-flash caused task accuracy to plateau while judge scores rose — hacking invisible from reasoning content, eliminated by re-balancing (Section 4.1).
>
> **2. On W2: "A lot of synthetic data generation is driven with a human in the loop... selection/inspection of samples... design of synthetic data generation pipelines (prompts, sampling strategies, multi-stage approaches)."** This is an important point. In our framework, the reviewer's three examples map directly:
>
> - *"selection/inspection of samples"*: Acts as a filtering signal analogous to rejection sampling — humans inject $I(X;S \mid Z_t) > 0$ by selecting or rejecting samples based on quality criteria (Section 3.4).
> - *"design of... pipelines (prompts, sampling strategies)"*: Defines which output distinctions matter — i.e., specifies what Section 4.1 calls the "acceptable set" $\mathcal{A}$.
> - *"multi-stage approaches"*: Human involvement maintains information openness (Section 3.3): each human intervention is a signal not derived from the model. Anthropic's generator-evaluator harness (post-submission, 2026) automates this without a human in the loop — a fixed evaluator agent plays the external signal role.
>
> An empirical observation reinforces this. In ongoing work, we tasked an LLM with iteratively self-reviewing and self-editing a research paper without human intervention: it read the paper, wrote reviewer-style comments, edited accordingly, repeating for ~7–8 iterations. Each iteration's comments appeared reasonable, and the LLM's self-assessment improved steadily. However, human review of the final version revealed severe quality degradation. This is a direct manifestation of Section 3.2's closed loop: without an external signal (human review), the cycle could only lose information about the true quality target, even though internal metrics appeared to improve.
>
> **3. On Key Questions: "What is the take home message for a practitioner?"** Three concrete guidelines:
>
> (i) *Ensure information openness (Section 3.3)*: Every synthetic data pipeline must include at least one signal source not derived from the model being trained. If all feedback comes from the model itself, the loop is closed and degradation is expected.
>
> (ii) *Prefer high meta-level signals (Section 4.1)*: When choosing between fine-grained supervision (matching a reference token by token) and coarse supervision (binary correctness), prefer the coarser signal. It is more sample-efficient because each observation addresses the task-relevant distinction, rather than spending capacity on identifying one surface form among many acceptable ones.
>
> (iii) *Invest in signal diversity*: Section 4.2 shows that high-efficiency injection induces constraints that transfer across prompts, domains, and interaction formats — covering diverse task-relevant regions yields better returns than more examples in already-covered regions.
>
> **4. On W3: "RLVR used before it is defined."** Thank you — we will define RLVR at first use in revision.

---

> > ### Author Rebuttal · Reviewer_XBh4 · 2026-04-01
> >
> > Thanks for the rebuttal and the expanded explanations, if space available it would be great to expand on those topics in the paper itself.

---

### Official Review · Reviewer_JDB8 · 2026-03-10

**Soundness:** 3
**Presentation:** 3
**Significance:** 2
**Originality:** 2
**Overall Recommendation:** 4
**Confidence:** 3

**Summary:**

This paper attempts to explain the phenomenon of synthetic data being "sometimes effective and sometimes ineffective" in large language model training from an information theory perspective. Based on the Data Processing Inequality (DPI), the authors propose that synthetic data can only consistently yield benefits when the generation-training loop is "information-open." If the training process mainly relies on model-generated samples and lacks continuous external task-related signal injection, this closed loop will lead to information contraction and capability degradation. Building on this, the paper further proposes the concept of "information injection efficiency," arguing that efficient synthetic supervision typically relies on higher meta-level signals, such as the correctness of the validator definition, format constraints, and stable evaluation criteria, rather than instance-level superficial imitation. Overall, this paper attempts to provide a unified theoretical framework to explain the successes and failures of recent works on synthetic data, RFT, RLVR, and verifier-guided training.

**Compliance With Llm Reviewing Policy:**

Affirmed.

**Key Questions For Authors:**

see in Weaknesses

**Limitations:**

- **Limited operationalizability of the theory.**
  Although the paper introduces concepts such as *information injection efficiency* and *meta-level supervision*, these notions still lack clear and quantifiable definitions, making them difficult to directly use for practical method design.

- **Lack of boundary-condition analysis.**
  The paper emphasizes the importance of external signals, but does not sufficiently discuss whether the proposed framework still holds when the external signals themselves are low-quality, biased, or unstable.

**Strengths And Weaknesses:**

### **Strengths**

1. **The paper addresses an important and practically relevant problem.** Synthetic data has become increasingly important for improving the training and inference capabilities of large models, yet its effectiveness and applicability still lack a unified explanation. This makes the problem studied in this paper highly relevant.

2. **The paper provides a clear and insightful synthesis of existing methods.** It not only discusses the limitations of closed-loop self-training under the DPI framework, but also introduces the notions of meta-level supervision and information injection efficiency to explain differences in sample efficiency and generalization across synthetic pipelines. This perspective is useful for understanding why some methods transfer more effectively across tasks.

### **Weaknesses**

1. **The novelty is somewhat limited; the paper reads more like a theoretical consolidation of existing empirical observations than a substantially new theory or method.** The main contribution appears to be a unified conceptual explanation of known phenomena. For example, both the risks of closed-loop self-training and the benefits of external validators or environmental feedback have already been supported by prior work on synthetic data, self-training, RLVR, and verifier-guided training. The value of this paper lies more in theoretical synthesis and clarification than in a strong original breakthrough.

2. **The proposed criterion is reasonable, but currently functions more as a post-hoc explanation than as actionable guidance.** The claim that external signal injection is crucial for effective synthetic data is convincing. However, many existing high-quality synthesis pipelines already rely on seed data, teacher models, external verifiers, executable feedback, or manual rules. As a result, while the proposed “information-open” criterion summarizes existing successful practices well, it remains unclear what genuinely new guidance it offers for designing future synthetic data pipelines. In other words, the paper explains why these methods have worked, but does not fully show how this theory can be used to systematically design better methods.

3. **The paper lacks sufficient experimental or extrapolative evidence to support the significance of its theoretical claims.** At present, the argument is mainly based on conceptual analysis and case summarization. To make the framework more convincing, the authors should provide more direct validation, for example:

    - controlling whether external signals are present to compare information-open and information-closed settings;

    - varying the stability, strength, or granularity of external signals to test their relationship with information injection efficiency;

    - comparing high-meta-level and low-meta-level supervision in terms of sample efficiency and OOD generalization.

---

> ### Author Rebuttal · Authors · 2026-03-27
>
> We thank the reviewer for the detailed and well-structured assessment.
>
> **1. On W1: "a theoretical consolidation of existing empirical observations rather than a substantially new theory."** We appreciate this characterization and would like to clarify the paper's new analytical content:
>
> (a) The per-iteration bound $I(X;Z_{t+1}) \le I(X;Z_t) + I(X;S \mid Z_t)$ (Section 3.4) is, to our knowledge, the first to formally quantify the information contribution of an external signal in iterative synthetic training. This is a derived result, not a summary of known phenomena.
>
> (b) The meta-level efficiency analysis (Section 4.1) provides an explicit entropy calculation showing *why* high meta-level signals are more sample-efficient — not a restatement of "RLVR works better than SFT," but an information-theoretic explanation through the structure of the supervision signal.
>
> (c) The framework connects previously separate phenomena — model collapse, RLVR generalization, reasoning content compression cross-domain transfer, verifier-guided discovery — under a single principle. We believe unification is a contribution when it reveals shared structure invisible from any single phenomenon.
>
> **2. On W2: "functions more as a post-hoc explanation than as actionable guidance."** The framework yields two specific, non-obvious design principles:
>
> - *Signal design (Section 4.1)*: Prefer supervision at the coarsest task-relevant granularity. The entropy calculation shows that a noisy binary verifier can be more efficient than clean reference data — this is not an obvious design choice, and it follows from the meta-level analysis rather than from existing empirical practice.
> - *Signal stability (Sections 3.2 + 3.3)*: Section 3.3 requires that external signals "not freely co-move with the current model"; if they do, Section 3.2's closed-loop monotonicity applies. In ongoing work, an LLM judge trained without external calibration drifted progressively — criteria shifted even as outputs appeared stable. Anthropic's generator-evaluator harness (published after our January 2026 submission) independently validates the fix: keeping the evaluator fixed prevents co-movement and yields large gains over single-agent approaches.
>
> **3. On W3: "lacks sufficient experimental or extrapolative evidence."** The reviewer's three suggestions are well-aligned with our framework. We supplement with evidence from ongoing work addressing each:
>
> - *Open vs. closed (suggestion 1)*: An instruction-following judge trained iteratively without external calibration showed progressive criteria drift — instantiating Section 3.2's closed-loop degradation. Anthropic's harness (post-submission, 2026) validates the fix: a fixed evaluator prevents co-movement.
> - *Varying stability/granularity (suggestion 2)*: We trained a judge for math/code/logic with all positives from Gemini 3 Pro (longer outputs) and negatives from MiMo-V2-flash (shorter outputs). The judge learned a length/style shortcut invisible from reasoning content — task accuracy plateaued while judge scores rose. Re-balancing eliminated the hacking and restored accuracy growth. Section 4.1 predicts exactly this: the coarser spurious signal (length/style) dominates over the intended signal (correctness).
> - *High vs. low meta-level (suggestion 3)*: Using gpt-oss-120b as a reward model in rubric-based RL for instruction following, we observed only 24% sample-level consistency across 16 independent scorings of 1,000 (prompt, response) pairs. Despite this low stability, RL converged to strong out-of-distribution (OOD) generalization — consistent with Section 3.4: random disagreements cancel across gradient updates and do not bias learning.
>
> **4. On Limitations: "information injection efficiency and meta-level supervision still lack clear and quantifiable definitions."** We respectfully note that Section 4.1 provides explicit quantitative definitions: information injection efficiency is $\Delta = I(Y;S \mid Q)$ — the bits of uncertainty reduction per supervised interaction. Meta-level is quantified by the entropy calculation: a high meta-level signal ($S_{\mathrm{high}} = \mathbf{1}[y \in \mathcal{A}]$) injects $\approx 1$ bit per observation, while a low meta-level signal ($S_{\mathrm{low}} = \mathbf{1}[y = y^\star]$) injects $\approx h_2(1/M) \to 0$ as $M$ grows. These are computed quantities, not qualitative labels.
>
> **5. On "boundary-condition analysis... low-quality, biased, or unstable."** The experiments in Point 3 directly address this: random noise (low-quality) is tolerable (Section 3.4); systematic bias is catastrophic (Section 4.1); instability (drifting judge) re-closes the loop (Section 3.2). We will expand this discussion in revision.

---

> > ### Author Rebuttal · Reviewer_JDB8 · 2026-04-05
> >
> > Thank you for your reply, my problem has been solved.

---

> > > ### Author Response · Authors · 2026-04-05
> > >
> > > Thank you so much for taking the time to read our rebuttal and for kindly acknowledging that your concerns have been fully resolved — we truly appreciate your effort and constructive feedback throughout this process.
> > >
> > > We noticed that the acknowledgement form suggests "please consider adjusting your score accordingly" for fully resolved cases. We wanted to gently bring this to your attention in case the score update was simply overlooked. Of course, we completely respect your judgment and understand if you feel the current score already reflects your overall assessment of the paper.
> > >
> > > Thank you again for your valuable time and support.

---

### Official Review · Reviewer_PXUB · 2026-03-16

**Soundness:** 3
**Presentation:** 4
**Significance:** 3
**Originality:** 3
**Overall Recommendation:** 4
**Confidence:** 4

**Summary:**

This paper seeks to explain how (and when) synthetic data contributes to performance by studying the injection of such data through an information theoretic lens. The key contribution claimed is that, with the synthetic data step added, is the information loop open or closed? If open, improements can be seen. If closed, not so much.

**Compliance With Llm Reviewing Policy:**

Affirmed.

**Key Questions For Authors:**

- Please see weaknesses above.
- If you could add one simple experiment, plot to illustrate this idea what would it be? (Even if the expt setting is too basic or well studied)
- There is a lot of mapping to other work going on (eg Sutton's Bitter Lesson) but some of this appear more motivational than concrete. It would be good if you can explain how to go about testing your theory going forward.

**Limitations:**

yes

**Strengths And Weaknesses:**

Key strengths:
+ The paper brings together a lot of interesting concepts, such as data processing inequality, mutual information, and the ways in which syn data is introduced to explain how and when synthetic data helps
+ They provide a good framework to cluster relevant work (but I am not sure this covers all related work).
+ Good attempt at casting RFT and RLVR in their framework
+ There is a good amount of philosophisizng going on (which I like - this paper can be a great position paper)

Weaknesses
- The main concern with this paper is that it is not falsifiable. i.e., how can somebody argue that the paper's arguments are incorrect? While the parts about DPI, Bayesian network modeling of what goes in an LLM tuning/adaptation process are good, the idea of how "high information injection" happens is quite handwavy.
- No experimental results. I agree that expt results for such an ambitious idea might not be feasible but the authors could have parameterized one type of LLM fine-tuning/adaptation process, created different percentages of synthetic dta infusion, and show that the theory has explanatory power. Even with the above the theory will not be predictive.

---

> ### Author Rebuttal · Authors · 2026-03-27
>
> We thank the reviewer for the encouraging assessment and the thoughtful questions.
>
> **1. On W1: "not falsifiable" and "the idea of how 'high information injection' happens is quite handwavy."** We appreciate this concern. The reviewer rightly notes that "the parts about DPI, Bayesian network modeling... are good"; our response focuses on making the "high information injection" part equally precise. Section 4.1 provides an explicit entropy calculation: for $M$ equally acceptable outputs, high meta-level feedback ($S_{\mathrm{high}} = \mathbf{1}[y \in \mathcal{A}]$) reduces ~1 bit per observation, while low meta-level feedback ($S_{\mathrm{low}} = \mathbf{1}[y = y^\star]$) yields $\approx h_2(1/M) \to 0$ as $M$ grows. This is a quantitative, testable claim.
>
> A concrete falsifiable test follows directly: take the same training data (prompts + reference solutions with reasoning traces). Condition (A): SFT on full references — the model must match the specific trace (low meta-level). Condition (B): discard reasoning traces, keep only final-answer correctness, run RL — the model reaches the correct answer by any path (high meta-level). Section 4.1 predicts (B) generalizes better despite using strictly less information, because it avoids spending capacity on within-class distinctions ($M$ valid reasoning paths). This is non-trivial: richer supervision is predicted to generalize *worse*. The gap should widen as $M$ increases.
>
> **2. On W2: "the authors could have parameterized one type of LLM fine-tuning/adaptation process... and show that the theory has explanatory power."** We supplement with three observations from ongoing work:
>
> *Observation 1 (random noise is benign — Section 3.4).* In rubric-based RL for instruction following, we used gpt-oss-120b as the reward model. Scoring 1,000 (prompt, response) pairs 16 times each, full agreement occurred in only 24% of cases (GPT-5.1 on CL-Bench (Dou et al., arXiv:2602.03587) shows similar rates). Despite this noise, RL converged to strong out-of-distribution generalization — consistent with Section 3.4: random disagreements cancel across gradient updates.
>
> *Observation 2 (systematic bias is catastrophic — Section 4.1).* Training a judge for math/code/logic where all positives came from Gemini 3 Pro (longer outputs) and all negatives from MiMo-V2-flash (shorter outputs), the judge learned a length/style shortcut invisible from its reasoning content: task accuracy plateaued while judge scores kept rising. Re-balancing sources eliminated the hacking and restored accuracy growth. In Section 4.1's terms, length/style is a coarser signal than correctness, so the model converged to it preferentially.
>
> *Observation 3 (closed-loop drift — Section 3.2).* An instruction-following judge trained iteratively without external calibration showed progressively decreasing acceptance rates — not because outputs worsened, but because criteria drifted without correction. This directly instantiates Section 3.2's closed-loop degradation.
>
> **3. On Key Question: "If you could add one simple experiment, plot to illustrate this idea what would it be?"**
>
> The most direct experiment: take a standard math benchmark (e.g., MATH). Split into train and held-out test sets. Train two models from the same checkpoint on the same training prompts:
>
> - Condition A (low meta-level): SFT on reference solutions with full reasoning traces — the model must reproduce the specific trace.
> - Condition B (high meta-level): discard reasoning traces, train with RL using binary verifier (correct/incorrect) — the model reaches the correct answer by any path.
>
> Plot: out-of-distribution (OOD) accuracy (e.g., on physics or coding problems not seen in training) vs. number of training samples. Section 4.1 predicts Condition B generalizes better, because binary correctness does not spend capacity distinguishing among the $M$ valid reasoning paths — it only separates correct from incorrect. The efficiency gap should widen as the diversity of valid solutions $M$ increases. This prediction is non-trivial: richer supervision (Condition A) is predicted to generalize *worse*.
>
> **4. On "mapping to other work (eg Sutton's Bitter Lesson) but some of this appear more motivational than concrete."** Fair point. The Bitter Lesson connection is in the Discussion section and is intended as broader contextualization. The concrete, testable content is in Sections 3–4: the per-iteration bound (Section 3.4) and the entropy calculation (Section 4.1) are formal results, not analogies.
>
> **5. On "how to go about testing your theory going forward."** Beyond the experiment in Point 3: (a) vary meta-level on the same task (binary correctness → step-level scoring → token-level imitation) and measure sample efficiency; (b) vary solution diversity $M$ to test whether the efficiency gap widens as Section 4.1 predicts.

---

> > ### Author Rebuttal · Reviewer_PXUB · 2026-04-02
> >
> > Thank you for the response and the updated experiments/proposals for experiments.

---

> > > ### Author Response · Authors · 2026-04-03
> > >
> > > Thank you very much for reading our rebuttal carefully and for acknowledging that your concerns have been fully resolved. Your constructive feedback has been very helpful in improving our paper.
> > >
> > > We noticed that the acknowledgement form mentions "please consider adjusting your score accordingly" for fully resolved cases, so we wanted to gently check whether you might consider revisiting the review score. We completely understand if you feel the current score already captures your overall assessment — we just wanted to make sure nothing was overlooked.
> > >
> > > Thank you again for your time and effort.

---

### Official Review · Reviewer_zDQt · 2026-03-22

**Soundness:** 1
**Presentation:** 2
**Significance:** 2
**Originality:** 2
**Overall Recommendation:** 2
**Confidence:** 3

**Summary:**

This paper study on when synthetic data can be beneficial and when are not.  The authors introduce the concept of data processing inequality(DPI), which attempts to explain the inconsistency of synthetic data.
They state that meta-level information in synthetic data generation play an decisive role in its utility.

**Compliance With Llm Reviewing Policy:**

Affirmed.

**Key Questions For Authors:**

Please see the weakness.

**Limitations:**

There is no literature review or related work section. For example, the published paper is highly related to this study: "Towards a Theoretical Understanding of Synthetic Data in LLM Post-Training: A Reverse-Bottleneck Perspective".

**Strengths And Weaknesses:**

Strengths:
- This paper has a clear problem statement in the beginning that makes it easy for readers to follow.
- The topic is timely, under the condition that many debates exist on the usefulness of the synthetic data.

Weaknesses:
- Even though the authors introduce a new concept of DPI and use mutual information and Markov chains to form their theoretical analysis, the whole insight is not new, not beyond the previous study, such as "self-Consuming Generative Models Go MAD" and "AI models collapse when trained on recursively generated data".
- The whole paper is full of assumption-like statements and misses rigorous proofs, for example, the gradient study in RLVR. More importantly,  it lacks experimental validations (either synthetic or real data), making readers less convinced.
-  The authors state that they provide an operational guide for synthetic data generation, but no such information is included in the paper except for explaining existing approaches in practice.
- This paper seems like "drawing the target around the arrow" to me. According to its current manuscript, it's more like a statement paper rather than fitting the main track of ICML.

---

> ### Author Rebuttal · Authors · 2026-03-27
>
> We thank the reviewer and address each concern below.
>
> **1. On W1: "the authors introduce a new concept of DPI" and "not new, not beyond" model collapse papers.** First, a clarification: DPI is a classical theorem in information theory (Cover & Thomas, 2006). We do not introduce it; we use it as an analytical tool. Second, the cited model collapse papers (Shumailov et al. (arXiv:2305.17493); Alemohammad et al. (arXiv:2307.01879)) establish that closed-loop self-training degrades — a finding we acknowledge in Section 3.2. Our paper makes three contributions absent from that literature:
>
> (a) Section 3.4 derives $I(X;Z_{t+1}) \le I(X;Z_t) + I(X;S \mid Z_t)$, quantifying *how much* an external signal can contribute per iteration. Model collapse diagnoses failure; this bound characterizes the *mechanism for success*.
>
> (b) Section 4.1 introduces meta-level efficiency with an explicit entropy calculation: high meta-level feedback (binary correctness) reduces ~1 bit per observation; low meta-level (matching one reference among $M$ acceptable solutions) yields $\approx h_2(1/M) \to 0$ as $M$ grows.
>
> (c) Section 3.5 unifies RFT and RLVR, showing how the same external signal enters through different mechanisms.
>
> We would appreciate identification of which of these results appear in the cited papers.
>
> **2. On W2: "full of assumption-like statements and misses rigorous proofs, for example, the gradient study in RLVR."** Sections 3.1–3.4 provide complete derivations: Markov chain formalization, DPI via chain rule, closed-loop monotonicity, and the refined bound with external signals. Section 4.1 gives an explicit entropy calculation. We note a significant divergence in soundness assessments across reviewers (1 to 4) and would appreciate specific identification of which derivations are considered unsound.
>
> **3. On W3: "no operational guide... except for explaining existing approaches."** The framework yields concrete non-obvious predictions beyond existing practices.
>
> *Signal design (Section 4.1)*: a noisy binary verifier can be *more* sample-efficient than clean reference data when many acceptable solutions $M$ exist — $h_2(1/M) \to 0$ as $M$ grows, making richer supervision *less* informative per sample.
>
> *Signal stability (Sections 3.2–3.3)*: signals that "co-move with the current model" close the loop and cause degradation regardless of current quality — keep the reward signal fixed or externally anchored. Anthropic's generator-evaluator harness (published after our January 2026 submission) independently validates this: a fixed evaluator, not co-moving with the generator, yields large gains over single-agent systems.
>
> **4. On W4: "drawing the target around the arrow... more like a statement paper."** We respectfully disagree. The per-iteration bound (Section 3.4) and the entropy calculation (Section 4.1) are derived results that yield falsifiable, non-trivial predictions — e.g., that SFT on full reference solutions should generalize *worse* than RL using only binary correctness from the same data, despite SFT using strictly richer information. This is not a post-hoc description of known outcomes; it is a counterintuitive prediction that follows from the formal analysis.
>
> **5. On Limitations: relationship to Gan & Liu (arXiv:2410.01720).** The frameworks differ: Gan & Liu treat the model as the information source ("reverse bottleneck"); we identify the external signal $S$ as the source — the model provides *exploration*, $S$ provides *selection pressure* (Section 3.4). Their analysis is one-shot; ours covers iterative dynamics. We will discuss this in revision.
>
> **6. Empirical observations from ongoing work** (addressing W2: "lacks experimental validations"):
>
> (i) *Random noise is benign (Section 3.4).* In rubric-based RL for instruction following, we used gpt-oss-120b as the reward model. Scoring 1,000 (prompt, response) pairs 16 times each, full agreement occurred in only 24% of cases (GPT-5.1 on CL-Bench (Dou et al., arXiv:2602.03587) shows similar rates). Despite this, RL training produced strong out-of-distribution generalization — random disagreements cancel across gradient updates.
>
> (ii) *Systematic bias is catastrophic (Section 4.1).* Training a judge for math/code/logic where all positives came from Gemini 3 Pro (longer outputs) and all negatives from MiMo-V2-flash (shorter outputs), the judge learned a length/style shortcut invisible from its reasoning content: task accuracy plateaued while judge scores kept rising. Re-balancing sources eliminated the hacking and restored accuracy growth. Length/style is a coarser signal than correctness, so the model converged to it preferentially.

---

> > ### Author Rebuttal · Reviewer_zDQt · 2026-04-03
> >
> > Thank you for the response.
> >
> > Even though the authors provide some preliminary results from ongoing work, concerns regarding missing empirical validation remain according to the current manuscript. Therefore, the proposed theory is not yet fully convincing and appears to rely heavily on assumptions. Notably, Reviewer PXUB seems to share a similar perspective: this paper could be a position paper.

---

### Decision · Program_Chairs · 2026-04-30

**Decision:**

Accept (regular)

**Comment:**

Though the reviews are mixed, the majority of reviewers are leaning towards weak acceptance, emphasizing that … The reviewers had mixed feelings about the "philosophizing" in the paper, with two reviewers emphasizing that it may be better suited as a position paper, one emphasizing that the lack of falsifiability of the claims is a major weakness, and pointing out that some of the proposed dimensions were quite high-level and "handwavey." Several reviewers emphasize that the paper would be much stronger with experimental justification that could back up the claims made. The ACs agree that the contributions of the paper have cleared the bar for acceptance, but encourage the authors to include the updated clarifications captured in the rebuttal into the camera ready, and consider grounding some of their assumptions theoretically or experimentally to strengthen the work.